# A lunar core dynamo limited to the Moon's first ~140 million years

Tinghong Zhou [1], John A. Tarduno [1,2,3] ✉, Rory D. Cottrell [1], Clive R. Neal [4], Francis Nimmo [5], Eric G. Blackman [2,3] & Mauricio Ibañez-Mejia [6]

Single crystal paleointensity (SCP) reveals that the Moon lacked a long-lived core dynamo, though mysteries remain. An episodic dynamo, seemingly recorded by some Apollo basalts, is temporally and energetically problematic. We evaluate this enigma through study of ~3.7 billion-year-old (Ga) Apollo basalts 70035 and 75035. Whole rock analyses show unrealistically high nominal magnetizations, whereas SCP indicate null fields, illustrating that the former do not record an episodic dynamo. However, deep crustal magnetic anomalies might record an early lunar dynamo. SCP studies of 3.97 Ga Apollo breccia 61016 and 4.36 Ga ferroan anorthosite 60025 also yield null values, constraining any core dynamo to the Moon's first 140 million years. These findings suggest that traces of Earth's Hadean atmosphere, transferred to the Moon lacking a magnetosphere, could be trapped in the buried lunar regolith, presenting an exceptional target for future exploration.

A new interpretation of lunar magnetism calls for the lack of a long-lived magnetic field of internal origin for most of the Moon's history[1]. Prior models had evoked a past long-lived lunar dynamo, producing a surface field at times as strong or stronger than Earth's field[2] and spanning some 2 billion years[3]. These conclusions, based on whole rock paleointensity (WRP) studies of Apollo samples, are paradoxical. The lunar core lacked sufficient energy to produce such a sustained field[4], and time-correlative strong, long wavelength lunar magnetic crustal anomalies[5] that should result from the long-lived dynamo are missing. The veracity of the magnetic data from prior analyses of Apollo samples had long been questioned[6], but only recently has evidence arisen for the absence of a long-lived dynamo[1].

The new data come from single crystal paleointensity (SCP)[7] analyses of Apollo samples. Lunar whole rock samples have notoriously poor magnetic recording properties[6]. By measuring single silicate crystals rather than whole rocks, specimens with ideal single-domain-like magnetic minerals can be isolated[8], meeting requirements for robust field recording[9]. SCP measurements of feldspars and pyroxene crystals from mare basalts 12053, 12040, 12021, 71055, and 14053, ranging in age from 3.2 to 3.9 Ga, yield null magnetizations[1]. Moreover, cooling experiments in known fields demonstrated that the crystals could have recorded magnetic fields with high efficiency if they had been present. These data thus define a Moon without a long-lived dynamo, with salient implications for future exploration. For example, the corresponding absence of a long-lived lunar paleomagnetosphere heightens the possibility that components of Earth's Archean atmosphere were transported to the Moon via the geomagnetosphere, and could be preserved in buried lunar regoliths[1,10]. The intensity of early solar wind and the close proximity of the Moon to Earth increases the likelihood of this terrestrial-lunar transfer[1,11,12].

With the insight provided by the SCP values indicating null values, some WRP data using thermal and nonthermal measurements are compatible with zero ambient fields[1]. However, the origin of Earth-like, or stronger, field strengths reported from other Apollo WRP studies[2] using nonthermal methods remains a mystery. Compression of the solar wind by impacts generally produces amplifications too small to account for the nominal Apollo WRP values[13,14]. Impact charge separation can create fields thousands of microTeslas[15–17]. Magnetic minerals in whole rock cooling through their Curie temperature could be magnetized by such impact plasmas[1]. But it is unlikely that all the Apollo samples recording high apparent fields[2] were cooling through the Curie temperatures of their respective magnetic minerals at the time of impacts[2]. This suggests another magnetization mechanism and/or that nonthermal techniques may not always be accurate[6,9].

We note that the apparent Earth-like WRP values between ~3.9 and ~3.6 Ga, sometimes called the "high field epoch"[3], have further motivated a model for an episodic lunar dynamo[18]. Evans and Tikoo[18] suggested that downwelling diapirs could locally increase core-mantle boundary heat flow, providing enough power to drive a core dynamo that could create a 50 µT lunar surface field for a total duration of 400 kyr. But this duration

[1]Department of Earth and Environmental Sciences, University of Rochester, Rochester, NY, 14627, USA. [2]Department of Physics and Astronomy, University of Rochester, Rochester, NY, 14627, USA. [3]Laboratory for Laser Energetics, University of Rochester, Rochester, NY, 14623, USA. [4]Department of Civil Engineering and Geological Sciences, University of Notre Dame, Notre Dame, IN, 46556, USA. [5]Department of Earth and Planetary Sciences, University of California, Santa Cruz, CA, 95064, USA. [6]Department of Geosciences, University of Arizona, Tucson, AZ, 85721, USA. ✉e-mail: john.tarduno@rochester.edu

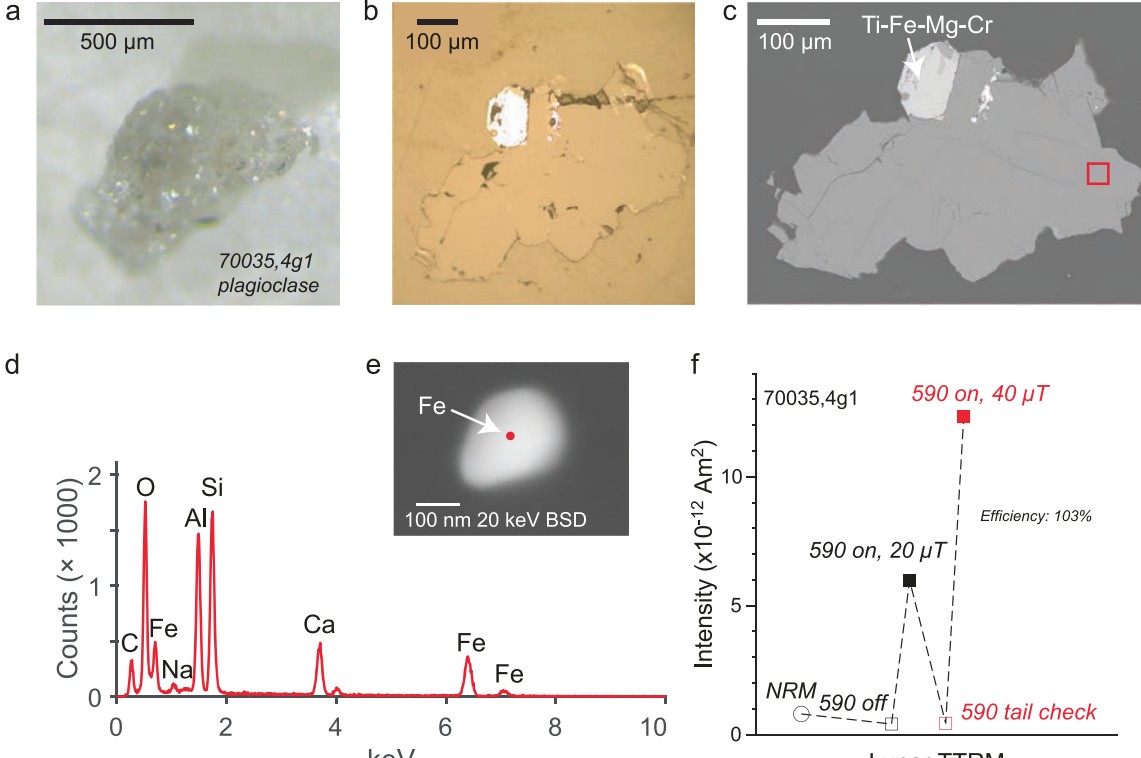

**Fig. 1 | Images and TRM experiment of Apollo sample 70035. a** Transmission light microscopy photo. **b** Reflected light image. **c** SEM backscatter (20 keV) image. Red box indicates the location of the analysis area (**d**, **e**). **d** EDS data with analysis spot highlighted by red dot in (**e**). **e** SEM backscatter (20 keV) image of particle analyzed with elements identified highlighted. **f** TRM experiment on the crystal imaged in (**a**–**e**). Intensity versus experiment steps. Efficiency is calculated from the field-on intensities of applied fields of 20 μT and 40 μT.

represents only 0.13% of the hypothesized high field epoch. Lunar rocks having high magnetizations imparted by this episodic dynamo would thus be rare, conflicting with their relatively common occurrence in the Apollo collection. Thus, an episodic dynamo explanation, whether by diapirs or impact-stirring of the core[18,19], exacerbates rather than solves the mystery of the high field values.

In contrast to the problematic long-lived or episodic dynamo hypotheses, satellite magnetic data suggest the Moon could have had a core dynamo in its very early history[5]. Weak magnetic anomalies of ~1 nT at 30 km altitude are particularly well-documented in the south polar region of the Moon[20], and are possibly representative of large areas (but not all) of the deep lunar crust[5]. However, the exact age of such a potential dynamo is unknown. We address both the origin of high apparent paleofields from WRP analyses and the age of any early lunar core dynamo through new analyses of Apollo samples.

## Results

### ~3.7 Ga Apollo high-Ti basalts

To investigate the origin of the high apparent paleofield values, we select two Apollo mare basalts for paired SCP and WRP analysis. We investigate Apollo 17 70035[21,22], a high-Ti mare basalt with several Rb-Sr[23,24] and Ar-Ar age analyses[25]. We recalculate the Rb-Sr isochrons using the latest [87]Rb decay constant calibration relative to the U-Pb system[26]. The new Rb-Sr isochrons (Supplementary Fig. 1) are 3812 ± 118 Ma for the data of Evensen et al.[23] and 3736 ± 114 Ma for that of Nyquist et al.[24]. Stettler et al.[25] measured two separate aliquots of 70035 for Ar-Ar. Using decay constants of Renne et al.[27] results in ages of 3656 ± 60 Ma and 3686 ± 51 Ma. Combining the four dates yields a weighted mean average of 3692 ± 34 Ma. The MSWD of 2.1 indicates the four values are in agreement within uncertainty.

We also investigate Apollo 17 75035, a high-Ti basalt[21] also the subject of prior geochronological studies[28,29], which we recalculate using recent decay constant and flux monitor data. The new Rb-Sr isochron using data of Murthy et al.[29] (Supplementary Fig. 1) yields 3818 ± 127 Ma. Recalculation of Ar-Ar whole rock data of Turner and Cadogan[28] yields 3734 ± 50 Ma, 3739 ± 40 Ma, and 3741 ± 40 Ma measured on plagioclase. The four ages yield a mean of 3741 ± 24 Ma. The MSWD of 0.5 again indicates agreement within uncertainty for all ages available from this sample.

### High-Ti basalts 70035, 75035, SCP results

We find that the natural remanent magnetizations (NRMs) of 70035 feldspar crystals are extremely weak, suggesting that their magnetic minerals cooled in the absence of a magnetic field ("Methods" section). After heating to 590 °C, a considerable portion of lunar magnetic carriers should remain blocked[1] and a magnetization should be observed if the feldspars had cooled in the presence of a lunar magnetic field. However, after demagnetization at 590 °C we find a magnetization indistinguishable from zero. Given this null magnetization state, the standard Thellier paleointensity approach is meaningless. Instead, we assess whether the crystals can record a magnetic field in accordance with magnetization theory[9] by the following procedure[1] ("Methods" section) (Fig. 1). First, we impart a partial thermoremanent magnetization (pTRM) at 590 °C in the presence of a 20 μT field. The sample is then demagnetized by heating to 590 °C, and the magnetization assessed to determine whether it returned to the null magnetization state. Next, a pTRM is imparted in a 40 μT field. The magnetizations of the two field strength pTRMs allow a determination of the recording efficiency[1]. We find the 70035 feldspars pass alteration checks ("Methods" section) have high efficiencies (103%; 96%, Fig. 1, Supplementary Fig. 2) and can record dynamo fields, but instead, they record zero magnetization levels. As a final check of the magnetization recording fidelity, we conduct scanning electron microscopy (SEM) and electron energy dispersive spectroscopy (EDS) analyses on the exact crystals used for SCP measurements (Fig. 1). These analyses confirm that the crystals contain magnetic carriers with

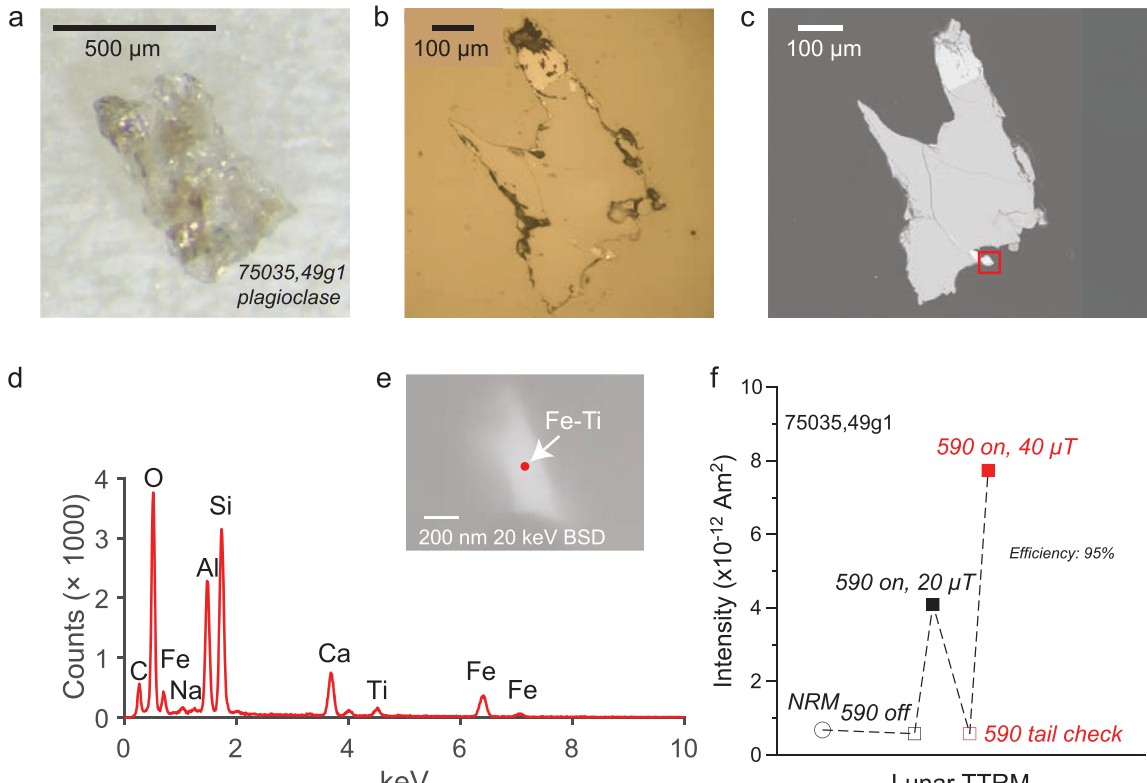

**Fig. 2 | Images and TRM experiment of Apollo sample 75035. a** Transmission light microscopy photo. **b** Reflected light image. **c** SEM backscatter (20 keV) image. Red box indicates the location of the analysis area (**d**, **e**). **d** EDS data with analysis spot highlighted by red dot in (**e**). **e** SEM backscatter (20 keV) image of particle analyzed with elements identified highlighted. **f** TRM experiment on the crystal imaged in (**a–e**). Intensity versus experiment steps. Efficiency is calculated from the field-on intensities of applied fields of 20 μT and 40 μT.

single-domain-like (SD) and/or single vortex (SV) grain sizes. Specifically, these include isolated Fe particles (Fig. 1e, Supplementary Fig. 3a–c h–j, l, m), Fe-Ti grains (Supplementary Fig. 3a, d), and Fe grains adjacent to Fe-S (Supplementary Fig. 3e–g).

The NRM of feldspars from 75035 also pass alteration checks and are essentially zero, where applied field experiments indicate the capability of the crystals to acquire fields with high efficiency (95%; 71%) (Fig. 2, Supplementary Fig. 2). SEM and EDS analyses further confirm the presence of minute inclusions within the size range of ideal magnetic recorders, namely SD-like or SV. These include Fe-Ti particles (Fig. 2e, Supplementary Fig. 4a, c, h, l) isolated Fe particles (Supplementary Fig. 4a, b), and Fe adjacent to minor Fe-S (Supplementary Fig. 4e–g).

### High-Ti basalts 70035, 75035, WRP results
Specimens from our 70035 sample show an irregular but a systematic demagnetization pattern after smoothing (Fig. 3a, Supplementary Fig. 5, "Methods" section). We use the REM' nonthermal technique (See "Methods" section for definition), applying and subsequently demagnetizing a saturation remanent magnetization (Fig. 3b, c), to estimate the paleofield strength. For one specimen, the component structure is complex; while a high coercivity direction can be isolated, the demagnetization of the saturation remanence is irregular, precluding a paleointensity estimate (Supplementary Fig. 5). For another, however (Fig. 3a–c), a distinct component can be isolated between 10 and 40 mT and this yields a nominal paleofield of 15.9 ± 2.8 μT.

Similarly, demagnetization data for 75035 show systematic trends after smoothing to reduce noise. For one specimen, the REM' data yield a nominal paleofield of 155.7 ± 31.7 μT isolated between 20 and 50 mT (Fig. 3d, "Methods" section). There is some nonlinearity in the saturation remanent demagnetization curve (Fig. 3e), possibly indicating a net magnetic anisotropy, motivating a consideration of additional specimens from 75035 (Fig. 3g–i). Another specimen again yielded systematic changes indicating multiple components of magnetization, with less nonlinearity displayed in the demagnetization of the saturation remanent magnetization curve (Fig. 3g–i). A very high coercivity component was isolated for this specimen between 100 and 120 mT which yields a nominal paleointensity of 259.9 ± 77.4 μT.

For completeness, we have conducted SEM analyses of the whole rock specimens used for the REM' paleointensity experiments. We find these contain relatively large (often 10 μm) multidomain (MD) Fe grains (Fig. 4), which are nonideal carriers.

### 3.97 and 4.36 Ga Apollo feldspars
To further assess the age limits on any early lunar core dynamo, we analyze components of two Apollo 16 breccias. Apollo 16 61016 is a dimict breccia with a reported U, Pu-[136]Xe age[30] of 3.97 ± 0.25 Ga. We also analyze Apollo 16 60025, a ferroan anorthosite with an age of 4360 ± 3 Ma based on [207]Pb-[206]Pb, [147]Sm-[143]Nd, and [146]Sm-[142]Nd isotopic systems[31].

### Dimict breccia 61016
We analyzed ~0.5 mm crystals of plagioclase (maskelynite). The NRM values are quite weak and after heating to 590 °C are weaker still, consistent with a null magnetization (Fig. 5, Supplementary Fig. 6). TRM experiments in two different fields pass alteration tests and show that the crystals have the ability to record the field at high efficiency (97%; 91%). SEM and EDS analyses show the presence of Fe particles (Fig. 5e, Supplementary Fig. 7a, b, d, e, g, h) and Fe-Ti grains (Supplementary Fig. 7j–m) within the size range of ideal SD/SV recorders (Fig. 5).

### Ferroan anorthosite 60025
We find that the NRM of feldspars from Apollo 60025 are extremely weak, both NRM values and after heating to 590 °C, consistent with null

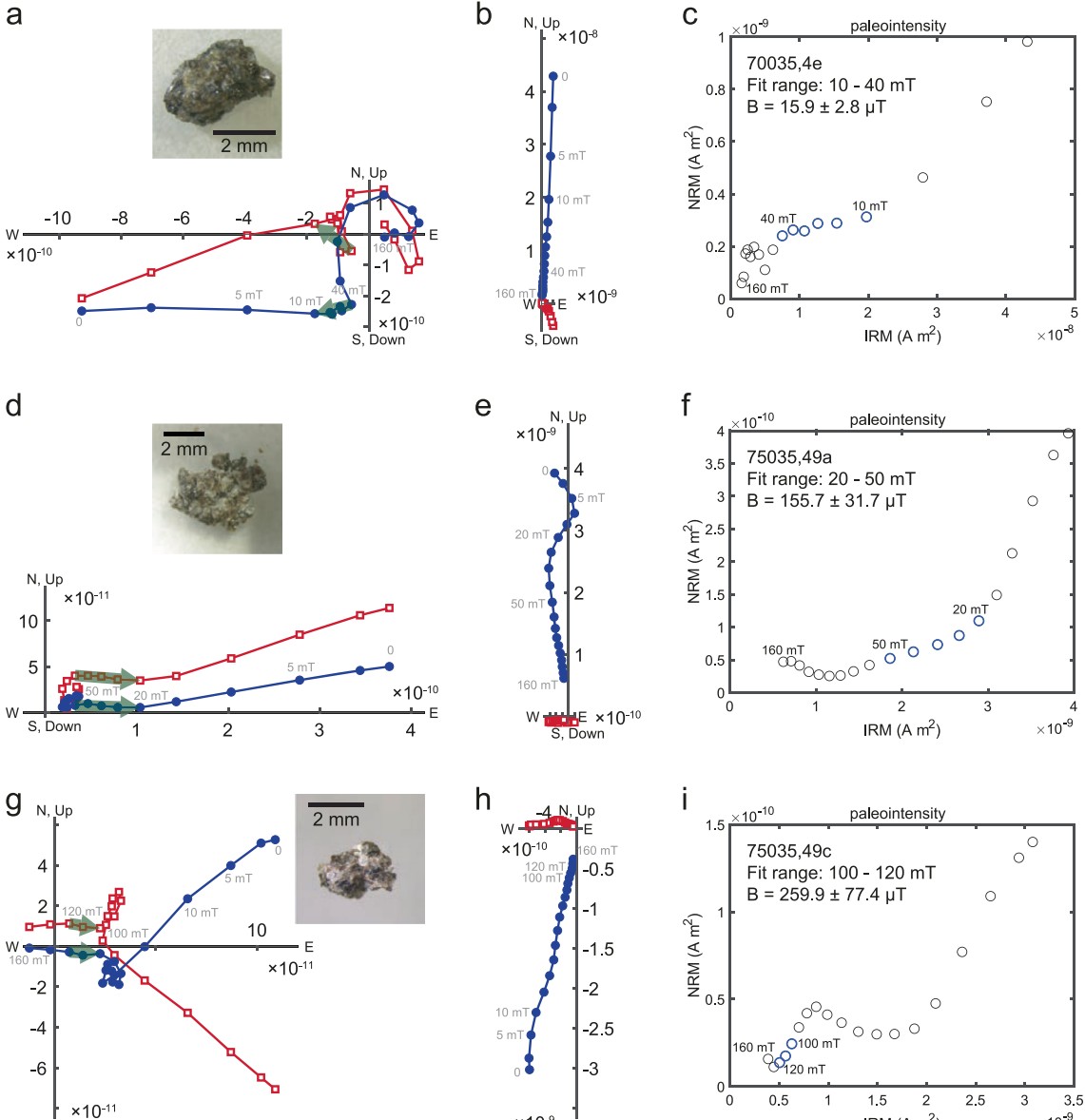

**Fig. 3 | REM' experiment for whole rock subsamples from Apollo samples 70035 and 75035. a** Orthogonal vector plot of AF demagnetization of NRM for 70035,4e (inset picture). The data presented are after two successive 3-point moving averages ("Methods" section). Demagnetization steps labeled in gray. Blue circles, horizontal projection; red squares, vertical projection; green arrows, vector corresponding to demagnetization range used in the paleointensity fit. **b** Orthogonal vector plot of AF demagnetization of a saturation remanent magnetization after 2 successive steps of 3-point moving average smoothing. Symbols as in (**a**). **c** NRM versus saturation remanent magnetization (circles). Demagnetization range used to calculate nominal paleointensity shown in blue. **d–f** Analyses for 75035,49a following conventions on (**a–c**) above. **g–i** Analyses for 75035,49c following conventions on (**a–c**) above.

magnetizations (Fig. 6). Applied TRMs ("Methods" section) again pass alteration checks (Fig. 6, Supplementary Fig. 6). Efficiencies are somewhat less than in the Apollo basalts or dimict breccia (87%; 66%) but still high enough to record ambient magnetic fields had they been present. This lower efficiency may be related to the composition of the carriers which are Fe-Ti (Fig. 6e, Supplementary Fig. 8a, b, e–g, h, i, l, m), Fe with Mg (Supplementary Fig. 8h, j) or Fe with both Mg and Ti (Supplementary Fig. 8a, c). The sizes and shapes, however, are consistent with SD/SV magnetic recording properties able to record paleofields for billions of years.

## Discussion

Our new results from 3.7 Ga high-Ti basalts 70035 and 75035 shed light on both the origin of apparent Earth-like magnetizations recorded by some Apollo whole rock samples[2] and the proposed, but temporally paradoxical, transient lunar dynamo[18]. The single silicate crystals from these basalts have magnetic inclusions with sizes consistent with ideal magnetic recording properties. These crystals also show high recording efficiencies, but record null ambient fields. In contrast, the bulk rocks contain nonideal MD magnetic particles, and yield strong yet variable magnetizations using nonthermal techniques (Fig. 7). A similar finding results from a comparison of SCP and WRP results for ~3.9 Ga Apollo 14 low-Ti/high-Al basalt 14053[32]. Single plagioclase and pyroxene yielded null magnetizations[1], whereby WRP analyses (discussed in ref. 1) yielded 20 μT using the nonthermal REM' method.

There are two reasons the WRP values may not be recording the true paleointensity values. First, nonthermal paleointensity methods may not yield reliable paleointensity data in these lunar whole rocks because they contain complex MD magnetic particles. Nonthermal techniques sometimes yield erroneous high values when applied to terrestrial basalts of known age[33]. The reason why nonthermal methods yield correct values on

**Fig. 4 | Scanning electron microscope images and EDS analyses for whole rock subsamples from Apollo samples 70035 and 75035 used for REM' experiment. a** Left: backscatter image (20 keV) from 70035,4d with elements identified; right: EDS data with collection spot highlighted by the red dot. **b** Left: backscatter image (20 keV) from 75035,49a with elements identified; right: EDS data with collection spot highlighted by the red dot.

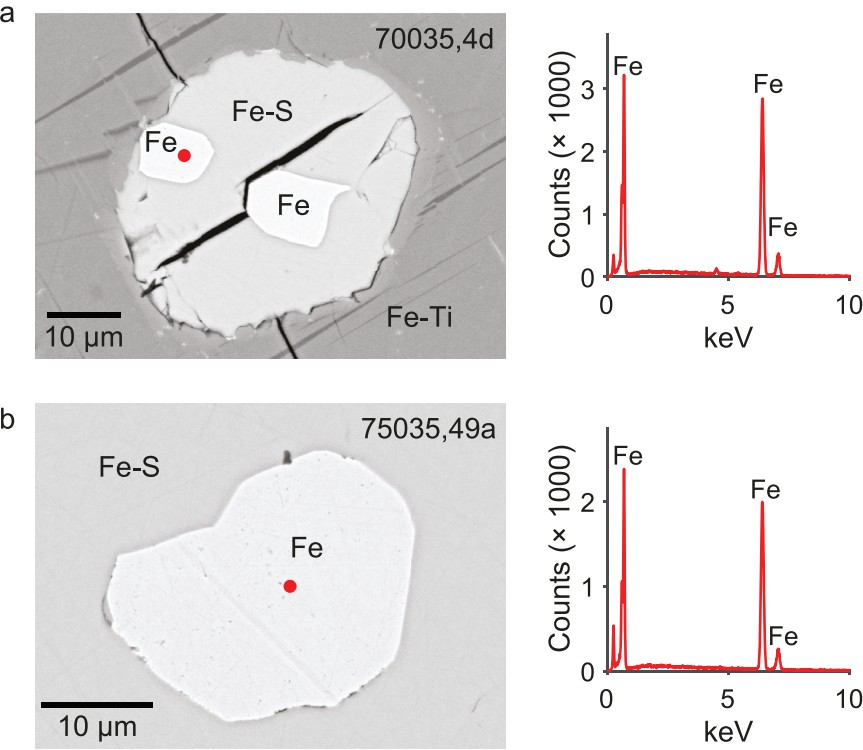

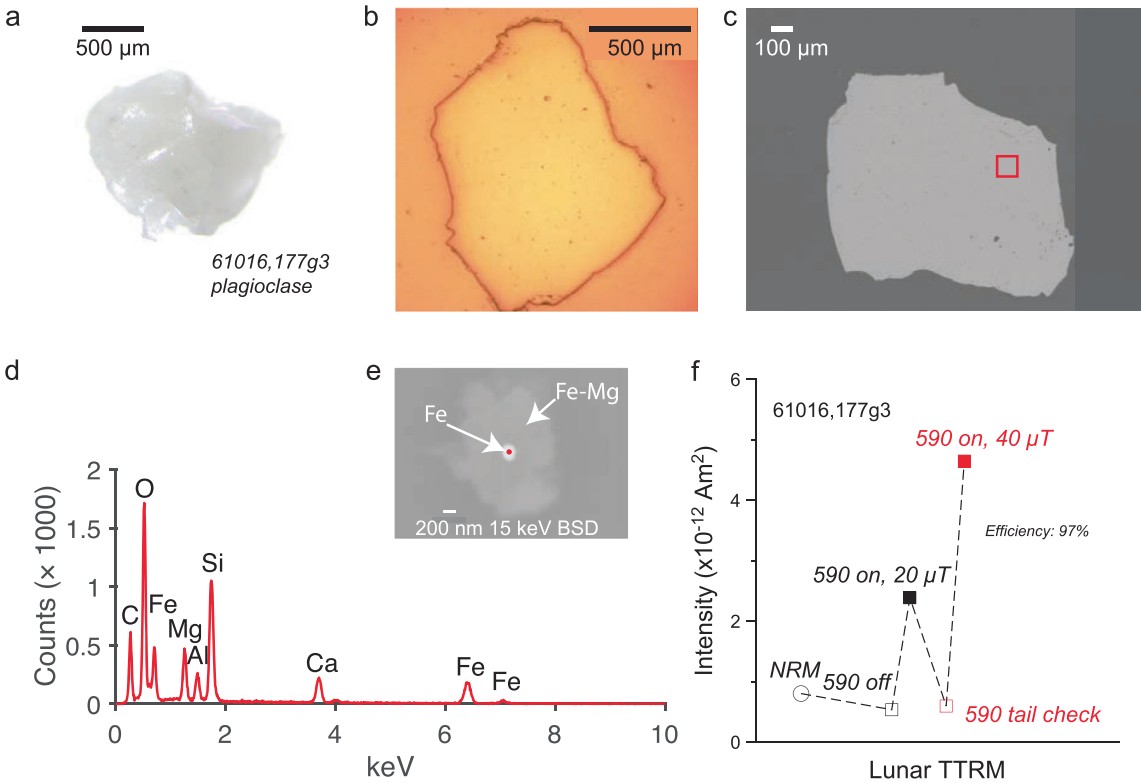

**Fig. 5 | Images and TRM experiment of Apollo sample 61016. a** Transmission light microscopy photo. **b** Reflected light image. **c** SEM backscatter (15 keV) image. Red box indicates the location of the analysis area (**d, e**). **d** EDS data with analysis spot highlighted by red dot in (**e**). **e** SEM backscatter (15 keV) image of particle analyzed with elements identified highlighted. **f** TRM experiment on the crystal imaged in (**a–e**). Intensity versus experiment steps. Efficiency is calculated from the field-on intensities of applied fields of 20 µT and 40 µT.

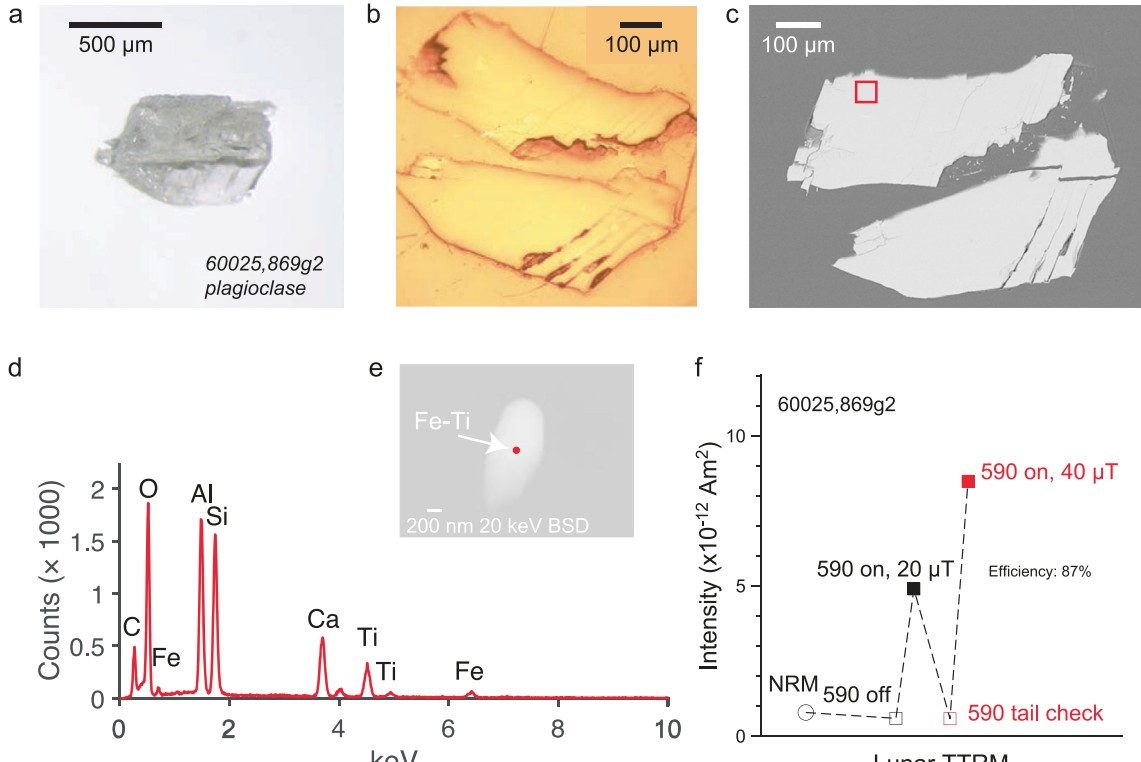

**Fig. 6 | Images and TRM experiment of Apollo sample 60025. a** Transmission light microscopy photo. **b** Reflected light image. **c** SEM backscatter (20 keV) image. Red box indicates the location of the analysis area (**d**, **e**). **d** EDS data with analysis spot highlighted by red dot in (**e**). **e** SEM backscatter (20 keV) image of particle analyzed with elements identified highlighted. **f** TRM experiment on the crystal imaged in (**a**–**e**). Intensity versus experiment steps. Efficiency is calculated from the field-on intensities of applied fields of 20 µT and 40 µT.

some terrestrial basalts but not others is unknown. But one common factor between the terrestrial and lunar basalts is the presence of nonideal MD magnetic particles. It is also possible that the more reduced composition of lunar magnetic carriers relative to those in terrestrial rocks may exacerbate anomalous response by MD grains in nonthermal paleointensity approaches. A second explanation relates to impacts. After rock crystallization, an impact can impart a shock remanent magnetization (SRM) with the magnetic field supplied by the attendant magnetized plasma[15–17]. The SRM would be preferentially recorded by the whole rock samples because the relatively large MD magnetic grains they host have domain walls that can move by shock, resulting in a net magnetization[9]. In contrast, the smaller SD-like magnetic grains in the single feldspar crystals would have higher coercivities and would be less likely to be magnetized by shock, preserving the true null field state during their cooling. SRM's have been considered previously[34], especially for 14053. The magnetizing fields are well within the range of impact plasmas and the charge-separation process[15–17]. We note that the explanations of the high apparent magnetizations from whole rocks are not mutually exclusive for explaining the entire record of magnetizations reported from Apollo samples.

The proposed episodic lunar magnetic field relies on the interpretation that high apparent WRP values[2] require a core dynamo[2,3,18]. Our observations indicate that the apparent high WRP values, whether they be nonthermal measurement artifacts or measures of shock and plasma fields, are not records of lunar core dynamo fields. Therefore, we conclude there is no reliable evidence for an episodic lunar dynamo from Apollo rocks.

The Moon lacks a magnetic field today, and therefore the null hypothesis is that it lacked one in the past, consistent with the limited energy to drive a dynamo in the relatively very small lunar core. Our new data do not reject this null hypothesis and instead further delineate the timeline of zero field measurements that provide evidence for the absence of a long-lived lunar dynamo (Fig. 7, "Methods" section, Supplementary Table 1). The observation of a null magnetization from Apollo 61016 indicates not only the lack of a dynamo at 3.97 Ga, but that any impact plasma magnetization had decayed before the sample cooled. The ultrafine magnetic minerals in the 61016 maskelynite also do not carry any subsequent shock magnetization, consistent with our results from 70035 to 75035.

The 4.36 Ga age of 60025 is ~100 million years older than the oldest age tentatively assigned to South Pole Aiken (SPA)[35] but the SPA age remains uncertain. Multiple hypotheses have been put forth to explain the formation of SPA[36] and its crustal anomalies[37], and until the age of SPA is better constrained we cannot determine whether these formed before or after the shutdown of a hypothetical early dynamo constrained by the 60025 SCP data.

Our data at 4.36 Ga sets a youngest bound on the age of a hypothetical early lunar dynamo. The oldest bound would be set by the formation age of the Moon itself, and for this, there remains debate. Representing this uncertainty, Halliday and Canup[38] bounded the lunar formation at 70–120 million years after the age of the Solar System, set at 4.5673 Ga. This suggests that a hypothetical dynamo could have been active in the first 87–137 million years after the formation of the Moon. A slightly older bound on the formation of the Moon is provided by the two-stage Hf-W model of core formation in Earth of 4.533 Ga, which provides the earliest time at which core formation in Earth can have ceased[39]. Since the last core formation event on Earth is thought to have been triggered by the Moon-forming impact, this model age also provides the earliest time at which the Moon can have formed. This yields a slightly longer duration of 170 million years for the hypothetical dynamo. But W isotope measurements of lunar samples suggest lunar differentiation occurred later than ~70 million years after the formation of the Solar System[38,40]. Therefore, we prefer a ~140 million years as the current best estimate of the upper bound on the duration of a hypothetical early dynamo.

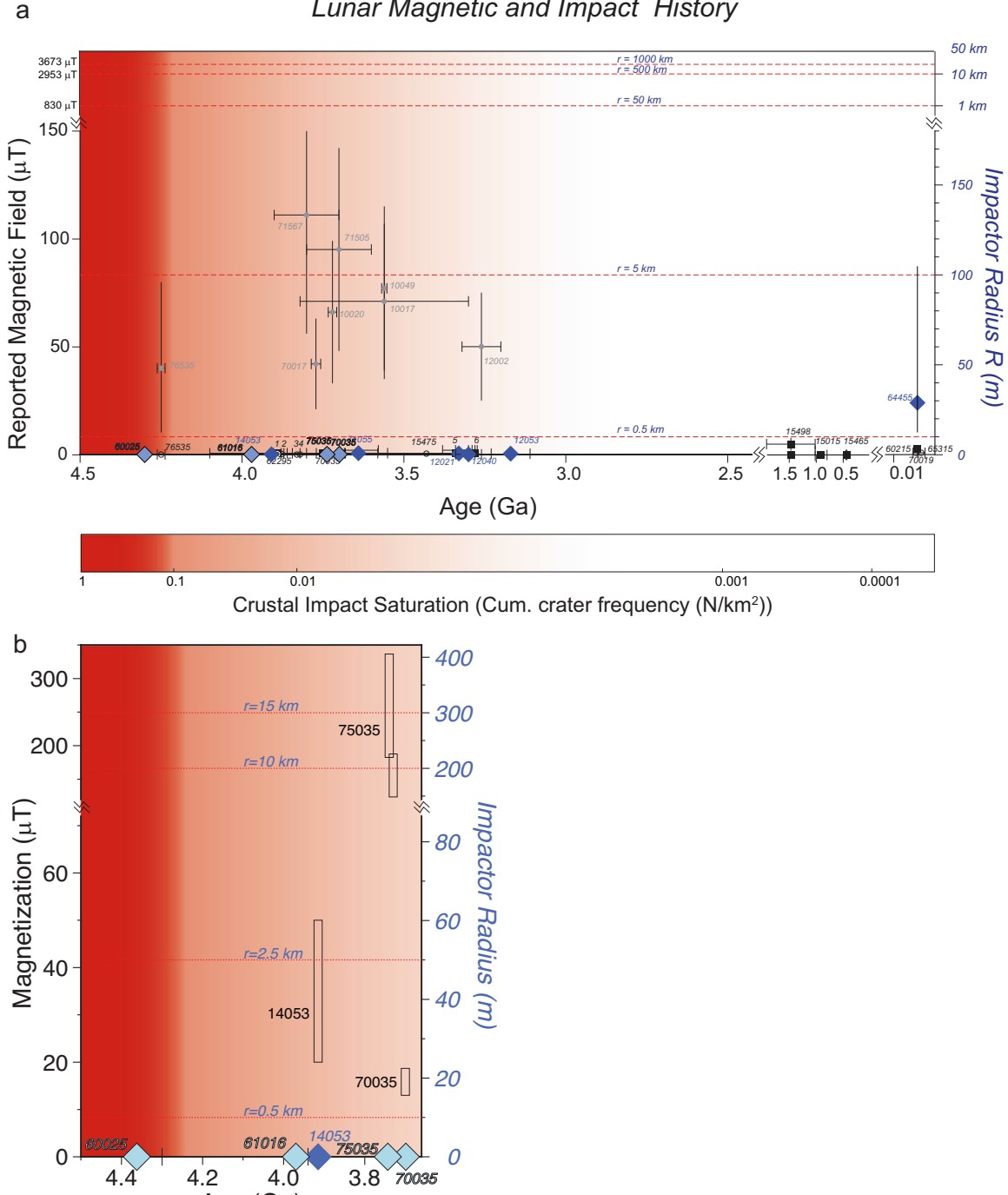

**Fig. 7 | Summary lunar magnetization and impact history. a** History after ref. 1 with new SCP values (light blue diamonds). Dark blue diamonds are other SCP values and values from glass (64455) from ref. 1. Black symbols are WRP data consistent with zero or impact magnetization (15498). Grey circles are WRP values using nonthermal techniques. Impactor radius after ref. 15 and crustal impact saturation from ref. 41, 42. **b** Expanded view of new data. Open rectangles are WRP values from nonthermal methods. Value for 14053 is from ref. 34. Data from 75035 and 70035 are reported here. Data for 75035 have been slightly offset to show individual ranges.

This finding has importance for the nature of the lunar core and crust. The deep crustal anomalies seen on the Moon hint at an internal magnetization process[5]. The short duration of any dynamo could help explain why these weak anomalies are not seen everywhere on the Moon. Magmatic intrusions after the shutoff of any dynamo could reset previously magnetized crust. We suggest that any further exploration for a past lunar dynamo using rock samples should concentrate on this early lunar history. Because of the impact saturation of the crust[1,41,42], this record may only be retrievable

from drilling or by analyzing samples exposed by impact excavation. Key questions that could be addressed are whether paleointensities are consistent with an early thermally or procession-driven dynamo[43,44] and, if present, how long this dynamo really lasted.

In the case of a precession-driven dynamo prior to 4.36 Ga, this mechanism only becomes important at a Moon-Earth separation >26–29 Earth radii ($R_\oplus$)[43], and shuts off at ~48 $R_\oplus$. Outwards migration to >26–29 $R_\oplus$ takes less than 100 Myr in most models[45], but the model time to reach 48

$R_\oplus$ is typically of the order of 1 Gyr[43]. If a precession-driven dynamo is responsible for the early lunar magnetic field, it implies sustained, rapid outward migration of the Moon. Alternatively, simple thermally-driven dynamos have little difficulty in sustaining a magnetic field for 140 Myr.[46]

Our findings are especially important for future exploration and studies of the lunar regolith. Without a long-lasting lunar paleomagnetosphere, solar wind volatiles can be implanted, and ancient regolith should contain relatively high abundances of $^3$He and other resources[1,47]. Today, elements from Earth's atmosphere are transported to the lunar surface when the Moon passes through the Earth's magnetosphere (magnetotail)[48] because of the lack of a lunar magnetosphere. The preferential occurrence of hematite on the lunar nearside and inferences on ancient oxygen transport to the Moon from Earth's atmosphere after the Great Oxidation Event (ca. 2.4 Ga)[49] are consistent with our findings. Transport might have been even more effective in the ancient Moon because of the smaller Moon-Earth separation[44]. Solar winds associated with the rapidly rotating young Sun were intense during the early Solar System[12], aiding the transport of Earth's early atmosphere to the Moon. Our new data suggest the lack of a lunar magnetosphere extended back in time before 4.0 Ga, and thus transport through a terrestrial magnetosphere could have occurred during the Hadean Eon of Earth[50–52]. The early composition of the terrestrial atmosphere, and the question over the balance of gasses that promoted greenhouse warming to avoid complete freezing over of the planet given the faint young Sun[12,53], remains a grand challenge question for terrestrial planet evolution. Recovery of older regolith (drilling, or sampling of impact crater walls) and return to Earth for analysis would be a pathway to obtaining geochemical data constraining Earth's Hadean atmosphere and tackling this grand challenge question.

## Methods

We separate single silicate crystals (feldspar) using non-magnetic tools for SCP analyses[7,8]. Silicate crystals analyzed in this investigation are approximately 0.5 mm in size. Bulk rock samples analyzed are approximate 3 mm in size.

Remanences are measured in the University of Rochester's magnetically shielded room (ambient field <200 nT). We use the ultra-sensitive WSGI 3-component DC SQUID magnetometer for single crystal remanence measurement and the 2G SQUID magnetometer for whole rock remanence measurement. For nonheating methods we use REM', (or ratio of equivalent magnetizations using derivatives), following ref. 54. REM' is calculated over a given AF demagnetization range as the derivative of NRM demagnetization relative to isothermal remanent magnetization demagnetization. A saturation remanent magnetization was applied using a 3 T field. To address the possibility of gyroremanence during AF demagnetizations, we follow the protocol presented by Finn and Coe[55]. Progressive AF demagnetization is accomplished by permuting the direction in which gyroremanence may be acquired. A three-step running mean for orthogonal components is used to calculate directions.

For thermal experiments, we use $CO_2$ laser techniques[56] which afford heating times more than an order of magnitude shorter than standard paleomagnetic ovens. Samples are heated and cooled rapidly in air; a controlled (reducing) atmosphere is not used because this can promote further reduction and the formation of new magnetic particles[1,57] (Supplementary Discussion).

TRM analyses follow that described in Tarduno et al.[1]. Repeat measurements are performed at every step. A zero magnetization is assigned if the nominal weak magnetization measured above the WSGI 3-component DC SQUID magnetometer's threshold yields inconsistent directions between multiple measurements. Alteration checks are performed following Tarduno et al.[50] and Tarduno et al.[1]. The difference between the field-off steps before and after the first field-on step is used to check for the presence of multidomain grains and potential alteration during heating. The linearity of TRM acquisition, interpreted as a measure of recording efficiency, is defined as $M_{590,40\mu T}/(M_{590,20\mu T} \times 2) \times 100\%$, where $M_{590,40\mu T}$ and $M_{590,20\mu T}$ are the magnetizations imparted applied fields of 40 and 20 µT, respectively.

We use non-magnetic materials, documented in multiple laboratories[51], to mount crystals. We investigate magnetic mineralogy using a Zeiss Auriga scanning electron microscope (SEM) with an energy dispersive x-ray analysis (EDS) at the University of Rochester Integrated Nanosystems Center. Weight percentage estimates from EDS data are used to identify Fe-Ti grains that differ from ilmenite.

In addition to age recalibrations for Apollo samples 70035 and 75035, we recalibrate ages for Apollo samples yielding SCP data reported in Tarduno et al.[1] using updated decay constants (see Supplementary Table 1).

## Data availability

Data presented in this paper are available at the following link: https://doi.org/10.6084/m9.figshare.24639291.

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

## Acknowledgements

We thank G. Kloc for sample preparation and Sean O'Neil for electron microscopy analyses. We thank K. Lawrence for the sample selection. This work was supported by NSF grant EAR1656348 and NASA grant 80NSSC19K0510 (to J.A.T.).

## Author contributions

J.A.T. conceived and supervised the project; T.Z. conducted SEM and nonthermal paleointensity analyses. R.D.C. conducted thermal paleointensity analyses. SEM and paleointensity data were analyzed by T.Z., R.D.C., and J.A.T. M.I.-M. contributed recalibration of radiometric ages, C.R.N. petrologic context, and F.N. and E.B. on dynamo and impact implications, respectively. J.A.T. and T.Z. wrote the manuscript with contributions for all the authors.

## Competing interests

The authors declare no competing interests.
