## [Transparent Peer Review file · Communications Earth & Environment]

A lunar core dynamo limited to the Moon's first ~140 million years

Corresponding Author: Dr John Tarduno

Due to technical limitations, attachments originally included by the reviewers as part of their assessment appear at the end of this file.

Version 0:

Decision Letter:

Dear Dr Tarduno,

Please allow us to apologise for the delay in sending a decision on your manuscript titled "A lunar core dynamo limited to the Moon's first 140 million years". It has now been seen by 3 reviewers, and we include their comments at the end of this message. They find your work of interest, but some important points are raised. We are interested in the possibility of publishing your study in Communications Earth & Environment, but would like to consider your responses to these concerns and assess a revised manuscript before we make a final decision on publication.

We therefore invite you to revise and resubmit your manuscript, along with a point-by-point response that takes into account the points raised. Please highlight all changes in the manuscript text file.

In particular, please ensure that the revised manuscript meets the following editorial thresholds:

* Quantify the typical single crystal palaeointensity and whole rock palaeointensity domain states to provide a thorough validation for your conclusions

* Provide a discussion and assessment of the robustness of the age determinations for the samples on which your conclusions are based

* Place your work clearly in the context of other relevant literature including that which has contrasting conclusions.

Please use the following link to submit your revised manuscript, point-by-point response to the referees' comments (which should be in a separate document to any cover letter), a tracked-changes version of the manuscript (as a PDF file) and the completed checklist:

Link Redacted

We hope to receive your revised paper within six weeks; please let us know if you aren't able to submit it within this time so that we can discuss how best to proceed. If we don't hear from you, and the revision process takes significantly longer, we may close your file. In this event, we will still be happy to reconsider your paper at a later date, as long as nothing similar has been accepted for publication at Communications Earth & Environment or published elsewhere in the meantime.

Please do not hesitate to contact us if you have any questions or would like to discuss these revisions further. We look forward to seeing the revised manuscript and thank you for the opportunity to review your work.

Best regards,

Joe Aslin

Deputy Editor,
Communications Earth & Environment
<https://www.nature.com/commsenv/>
Twitter: @CommsEarth

EDITORIAL POLICIES AND FORMATTING

Editorial Policy: [Policy requirements](https://www.nature.com/documents/nr-editorial-policy-checklist.pdf) (Download the link to your computer as a PDF.)

- Behavioural and social science
- Ecological, evolutionary & environmental sciences
- Life sciences

<https://www.nature.com/documents/nr-reporting-summary.zip>

Furthermore, please align your manuscript with our format requirements, which are summarized on the following checklist: [Communications Earth & Environment formatting checklist](https://www.nature.com/documents/commsj-phys-style-formatting-checklist-article.pdf)

and also in our style and formatting guide [Communications Earth & Environment formatting guide](https://www.nature.com/documents/commsj-phys-style-formatting-guide-accept.pdf) .

*** DATA: Communications Earth & Environment endorses the principles of the Enabling FAIR data project (<http://www.copdess.org/enabling-fair-data-project/>). We ask authors to make the data that support their conclusions available in permanent, publically accessible data repositories. (Please contact the editor if you are unable to make your data available).

All Communications Earth & Environment manuscripts must include a section titled "Data Availability" at the end of the Methods section or main text (if no Methods). More information on this policy, is available at <http://www.nature.com/authors/policies/data/data-availability-statements-data-citations.pdf>.

If a community resource is unavailable, data can be submitted to generalist repositories such as [figshare](https://figshare.com/) or [Dryad Digital Repository](http://datadryad.org/). Please provide a unique identifier for the data (for example a DOI or a permanent URL) in the data availability statement, if possible. If the repository does not provide identifiers, we encourage authors to supply the search terms that will return the data. For data that have been obtained from publically available sources, please provide a URL and the specific data product name in the data availability statement. Data with a DOI should be further cited in the methods reference section.

REVIEWER COMMENTS:

Reviewer #1 (Remarks to the Author):

See attachment

Reviewer #2 (Remarks to the Author):

Paleomagnetic analyses of silicate single crystals (SCP) are used to convincingly demonstrate that the Moon was lacking a core dynamo for the last ~4.36 Ga, contrary to the scenario suggested by whole rock measurements (WRP). The reason for the strong discrepancy between SCP and WRP paleointensity results is attributed to WRP artifacts or to whole rocks having acquired an impact magnetization in the transient magnetic field produced by the impact plasma, or both. The ability of SCP to record a planetary field and hold it during impacts and the opposite behavior of WRP is attributed to the ideal behavior and high paleomagnetic stability of single-domain (SD) and small pseudo-single-domain (PSD) magnetic inclusions in silicates, as opposed to the non-ideal behavior, low stability, and shock sensitivity of large multidomain (MD) crystals in the bulk rocks.

The difference between SCP and WRP results is shown in Figures 2 and 3. Given the pivotal role played by the domain state of remanence carriers in explaining the results of this paper and validating its conclusions, it would be desirable to quantify, at least roughly, the typical SCP and WRP domain states. In the case of SCP, the SD/PSD state is deducible indirectly from TEM, while no information is available for the WRP example. ARM/IRM would be a suitable indicator for both cases. In the case of WRP, IRM (Figure 3) is used as a term of comparison for NRM because of the REM' method of ref. 45. IRM is an incredibly poor laboratory analogue of TRM — despite AF magnetization curves of IRM being more similar to those of NRM in the case of MD carriers than ARM — because it activates high-energy magnetic states that are never accessed in weak fields. Because the maximum AF field is not a limiting factor in the WRP example of Figure 3, I suggest to add AF demagnetization results from ARM. A better linearity of NRM:ARM plots would strengthen the domain state problem of WRP, whereas similarities with NRM:IRM would point to a real (impact) magnetization acquired by the bulk rock.

Other comments:

Second-last paragraph of page 6: In statistical testing, the null hypothesis is the one to be rejected. Therefore, your null hypothesis would be that the moon had a core dynamo lasting longer than 140 Ma, and SCP rejects it.

Last paragraph before data availability: $M_{40\mu T}/(2*M_{20\mu T})$ expresses the linearity of the acquisition mechanism, rather than its efficiency, which is given by $M(B)/B$.

Ramon Egli

Reviewer #3 (Remarks to the Author):

Review of “A lunar core dynamo limited to the Moon’s first 140 million years”

This study presents a new palaeomagnetic appraisal of Apollo samples supporting a re-interpretation of the history of lunar magnetism. This re-interpretation is in-line with a recent publication (by some of the authors on the present study), in which it is claimed that the magnetization was likely acquired as a result of impacts rather than cooling in the presence of dynamo field.

Overall, the study is well written, clear, and its conclusions are supported by the palaeomagnetic experiments.

My comments are minor but I consider them nevertheless important.

1) The interpretation that the magnetization in Apollo samples was acquired as a result of impacts, in the absence of a core dynamo field, is contrary to the conclusions reached by Oran et al (Science Advances, 2020). Furthermore, it is also contrary to the study of Wieczorek et al (Science 2012), who argue that lunar magnetic anomalies are consistent with an impact scenario and magnetized in a core dynamo field. These two papers are conveniently omitted from the reference list. But what has changed to make these two studies incorrect or superseded? References 13 to 15 are given in support for the authors favoured scenario, but an explanation for why the conclusions of Oran et al and Wieczorek et al are invalid must also be provided.

2) In addition to the episodic lunar dynamo suggested by Evans and Tikoo, one can also note the possibility that impacts can also lead to episodic dynamos (e.g. Le Bars et al, Nature, 2011)

3) Given the broad readership of the journal, It would be helpful to indicate more precisely why previous paleomagnetic measurements (that supported an ancient lunar dynamo) are incorrect.

4) Typical model times to model time for the lunar orbit radius to reach 48 Earth radii is only about 1.5 Gyr, not “several Gyr”.

Communications Earth & Environment is committed to improving transparency in authorship. As part of our efforts in this direction, we are now requesting that all authors identified as 'corresponding author' create and link their Open Researcher and Contributor Identifier (ORCID) with their account on the Manuscript Tracking System prior to acceptance. ORCID helps the scientific community achieve unambiguous attribution of all scholarly contributions. You can create and link your ORCID from the home page of the Manuscript Tracking System by clicking on 'Modify my Springer Nature account' and following the instructions in the link below. Please also inform all co-authors that they can add their ORCID to their accounts and that they must do so prior to acceptance.
<https://www.springernature.com/gp/researchers/orcid/orcid-for-nature-research>

Author Rebuttal letter:

May 11, 2024
Response to Comments
Reviewer 1

Comment: This is a bold paper proposing that the lunar core dynamo is limited to the first 140 million years of Moon's history on the basis of single crystal paleointensity (SCP) studies on 4 samples (ages 3.7 Ga, 3.97 Ga and 4.36 Ga) returned by the Apollo mission. The authors build on earlier work (Tarduno et al, 2021, Sci Advances) that calls into question previous estimates of lunar paleointensity based on whole rock samples, using both thermal and non-thermal paleointensity methods. That paper revealed evidence that SCP results on 5 samples ranging in age from 3.2 - 3.9 Ga did not record a dynamo field despite apparent capacity to do so. A very young sample (2 Ma) does retain a remanence but this is interpreted as a result of an impact not a native dynamo record. Supplementary material in the current manuscript indicates the existence of 3 samples (Apollo 70019, 15498, and 64455) with evidence for impact magnetization. The new manuscript is based on the comparison of SCP and whole rock values from the 2 youngest of the 3.7-4.36 Ga samples, and these are the basalts, 70035 and 75035. The SCP produces a null magnetic field, and the high whole rock results are considered inconsistent and unrealistic. On this basis they accept all 4 of their null SCP results as affirmative evidence that any lunar dynamo was confined to the 140 My interval prior to 4.36 Ga. They speculate that absence of a lunar magnetosphere would allow transfer of Earth's Hadean magnetosphere to Moon and an associated record in the deeply buried lunar regolith. Geochemists might find this an interesting target for future study.

Response: We thank the reviewer for this assessment.

Comment: As I see it the current paper provides two stated advances over the 2021 work. Firstly, a direct comparison is made between SCP and whole rock paleointensity records on the same sample, reaffirming enthusiasm for the still challenging SCP experiments - the single crystals do have very weak magnetic signals. Secondly, the age of any putative dynamo is extended to that of the older samples, a 3.97 Ga breccia (61016) and 4.36 Ga anorthosite (60025). (No whole rock comparison is made on these samples.) The ages of these samples are cited as based on results from references 27 and 28. I have no expertise to judge the actual reliability of the 3.97 ± 0.25 Ga and 4.36 ± 0.03 Ga age assignments, but the assertion of a dynamo lasting no longer than 140 My is clearly completely dependent on the second of these results - and perhaps on current views of age of Moon. Is 4.53 Ga still a good estimate (<https://eos.org/articles/the-moon-is-even-older-than-scientists-thought>)? I appreciate that the ages have been recalibrated with new decay constants, but could there be better determinations with more recent techniques? I found the manuscript lacking in detail about general reliability of these ages. Overall this is an interesting study. It builds on SCP evidence supporting previous results that the lunar dynamo history is less complex than initially thought, and could be made significant with a more nuanced discussion of the reliability of the oldest age constraints.

1

Response: We thank the reviewer for these comments. The age bound - separating

times where a field is possible from when it is absent - is fixed by the age of Apollo 16 ferroan anorthosite 60025 at $4,360 \pm 3$ million years using three different chronometers (^{207}Pb - ^{208}Pb , ^{147}Sm - ^{143}Nd , and ^{146}Sm - ^{142}Nd ; Borg et al., 2011, Nature 477, 70-72). The principal issue raised is the duration over which the dynamo could have been present, and as the reviewer notes this is fixed by the age of the Moon. We selected a range of ages to address this uncertainty provided by Halliday and Canup (2023), but we did not describe this. We have now revised the manuscript accordingly.

Comment: "Minor points: Should Moon be capitalized as a proper noun when it is preceded by the?"

Response: Yes.

Comment: "Define REM."

Response: We now define REM in the Methods and emphasize links to the Methods in the revised main text.

2

Reviewer 2 (Ramon Egli)

Comment: "Paleomagnetic analyses of silicate single crystals (SCP) are used to convincingly demonstrate that the Moon was lacking a core dynamo for the last 4.36 Ga, contrary to the scenario suggested by whole rock measurements (WRP). The reason for the strong discrepancy between SCP and WRP paleointensity results is attributed to WRP artifacts or to whole rocks having acquired an impact magnetization in the transient magnetic field produced by the impact plasma, or both. The ability of SCP to record a planetary field and hold it during impacts and the opposite behavior of WRP is attributed to the ideal behavior and high paleomagnetic stability of single-domain (SD) and small pseudo-single-domain (PSD) magnetic inclusions in silicates, as opposed to the non-ideal behavior, low stability, and shock sensitivity of large multidomain (MD) crystals in the bulk rocks."

Response: We thank the reviewer for this assessment.

Comment: "The difference between SCP and WRP results is shown in Figures 2 and 3. Given the pivotal role played by the domain state of remanence carriers in explaining the results of this paper and validating its conclusions, it would be desirable to quantify, at least roughly, the typical SCP and WRP domain states. In the case of SCP, the SD/PSD state is deducible indirectly from TEM, while no information is available for the WRP example. ARM/IRM would be a suitable indicator for both cases. In the case of WRP, IRM (Figure 3) is used as a term of comparison for NRM because of the REM method of ref. 45. IRM is an incredibly poor laboratory analogue of TRM despite AF magnetization curves of IRM being more similar to those of NRM in the case of MD carriers than ARM because it activates high-energy magnetic states that are never accessed in weak fields. Because the maximum AF field is not a limiting factor in the WRP example of Figure 3, I suggest to add AF demagnetization results from ARM. A better linearity of NRM:ARM plots would strengthen the domain state problem of WRP, whereas similarities with NRM:IRM would point to a real (impact) magnetization acquired by the bulk rock."

Response: We appreciate the issue raised. We prefer to use SEM analyses to document magnetic sizes/domain state, and have accordingly conducted analyses on 70035 and 75035. As expected from prior work, our whole rock samples contain magnetic grains that are huge (e.g. $10 \mu\text{m}$) relative to those in the single crystals, and clearly in the multidomain state. We now include these analyses as an additional figure in the manuscript and thank the reviewer again for raising this.

Comment: "Second-last paragraph of page 6: In statistical testing, the null hypothesis is the one to be rejected. Therefore, your null hypothesis would be that the moon had a core dynamo lasting longer than 140 Ma, and SCP rejects it."

Response: We see the reviewer's point, but we prefer to state that for the Moon, the

3

null hypothesis is that there was no past dynamo because there is no dynamo today. Through testing, our SCP data cannot reject the null hypothesis. We have added this to the manuscript.

Comment: "Last paragraph before data availability: $M \propto B^2 / (2 * M \propto B)$ expresses the linearity of the acquisition mechanism, rather than its efficiency, which is given by $M(B)/B$."

Response: Fair enough. We have revised the text to note this.

4

Reviewer 3

Comment: "This study presents a new palaeomagnetic appraisal of Apollo samples supporting a re-interpretation of the history of lunar magnetism. This re-interpretation is in-line with a recent publication (by some of the authors on the present study), in which it is claimed that the magnetization was likely acquired as a result of impacts rather than cooling in the presence of dynamo field. Overall, the study is well written, clear, and its conclusions are supported by the palaeomagnetic experiments."

Response: We thank the reviewer for this assessment.

Comment: "My comments are minor but I consider them nevertheless important. 1) The interpretation that the magnetization in Apollo samples was acquired as a result of impacts, in the absence of a core dynamo field, is contrary to the conclusions reached by Oran et al (Science Advances, 2020). Furthermore, it is also contrary to the study of Wieczorek et al (Science 2012), who argue that lunar magnetic anomalies are consistent with an impact scenario and magnetized in a core dynamo field. These two papers are conveniently omitted from the reference list. But what has changed to make these two studies incorrect or superseded? References 13 to 15 are given in support for the authors favoured scenario, but an explanation for why the conclusions of Oran et al and Wieczorek et al are invalid must also be provided."

Response: We thank the review for the opportunity to clarify the potential relevance of these papers. As discussed in the Tarduno et al. (2021) work, Oran et al (2020) modeled the amplification of the relatively weak solar wind field by compression. They did not consider that the field could be much larger due to charge separation driven currents from impacts (Crawford, 2020), which is the process we consider here. We have now clarified this in the manuscript.

Our single crystal data directly constrain the presence/absence of a dynamo at a given age. In a very nice study, Wieczorek et al. (2012) modeled the formation of South Pole Akin (SPA) and its magnetic anomalies. But SPA has not been directly dated. There are indirect ages from Ar-Ar analyses of Apollo 17 sample 76535 at 4.25 Ga, but it is unclear whether 76255 came from SPA. Until SPA is dated, we will not know whether its anomalies are consistent with an early dynamo, or require another mechanism. Because of this unknown, we decided not to include a specific discussion in this manuscript, but we now include a brief discussion of SPA and cite the Wieczorek et al. (2012) paper.

Also, we now include a reference to a study by Li et al. (2020) on occurrence of hematite on the Moon that is consistent with our findings.

5

Comment: "2) In addition to the episodic lunar dynamo suggested by Evans and Tikoo, one can also note the possibility that impacts can also lead to episodic dynamos (e.g. Le Bars et al, Nature, 2011)"

Response: We have added this, but with the caveat that the fields predicted are in general very small and occur over relatively short time intervals.

Comment: 3) Given the broad readership of the journal, It would be helpful to indicate more precisely why previous paleomagnetic measurements (that supported an ancient lunar dynamo) are incorrect.

Response: We have added additional words to the abstract to better highlight the key domain state difference between whole rocks, the basis of prior work, and that of the single crystals (the new work and findings).

Comment: 4) Typical model times to model time for the lunar orbit radius to reach 48 Earth radii is only about 1.5 Gyr, not several Gyr.

Response: We feel Fig 5c of Korenaga (2023, Icarus) is a good summary. A value of 48 Earth radii can be reached after several hundred Myr to 2 Gyr. We have revised the manuscript to say that it takes of order 1 Gyr.

6

Version 1:

Decision Letter:

Dear Professor Tarduno,

Your manuscript titled "A lunar core dynamo limited to the Moon's first ~140 million years" has now been seen by our reviewers, whose comments appear below. In light of their advice we are delighted to say that we are happy, in principle, to publish a suitably revised version in *Communications Earth & Environment* under the open access CC BY license (Creative Commons Attribution v4.0 International License).

We therefore invite you to edit your manuscript to comply with our format requirements and to maximise the accessibility and therefore the impact of your work.

EDITORIAL REQUESTS:

****Please take care to match our formatting and policy requirements. We will check revised manuscript and return manuscripts that do not comply. Such requests will lead to delays. ****

SUBMISSION INFORMATION:

OPEN ACCESS:

Communications Earth & Environment is a fully open access journal. Articles are made freely accessible on publication under a [CC BY license](http://creativecommons.org/licenses/by/4.0) (Creative Commons Attribution 4.0 International License). This license allows maximum dissemination and re-use of open access materials and is preferred by many research funding bodies.

For further information about article processing charges, open access funding, and advice and support from Nature Research, please visit <https://www.nature.com/commsenv/article-processing-charges>

At acceptance, you will be provided with instructions for completing this CC BY license on behalf of all authors. This grants us the necessary permissions to publish your paper. Additionally, you will be asked to declare that all required third party

permissions have been obtained, and to provide billing information in order to pay the article-processing charge (APC).

Link Redacted

Best regards,

Joe Aslin

Deputy Editor,
Communications Earth & Environment
<https://www.nature.com/commsenv/>
Twitter: @CommsEarth

REVIEWERS' COMMENTS:

Reviewer #1 (Remarks to the Author):

I have no further comments on the revised manuscript

Reviewer #2 (Remarks to the Author):

My comments have been addressed adequately and I have no further ones.

Reviewer #3 (Remarks to the Author):

I thank the author for revising their manuscript.

I am happy with the revisions.

Regards

Review of COMMSENV-24-0254-T

A lunar core dynamo limited to the Moon's first 140 million years

by Zhou et alia

This is a bold paper proposing that the lunar core dynamo is limited to the first 140 million years of Moon's history on the basis of single crystal paleointensity (SCP) studies on 4 samples (ages ~ 3.7 Ga, 3.97 Ga and 4.36 Ga) returned by the Apollo mission. The authors build on earlier work (Tarduno et al, 2021, Sci Advances) that calls into question previous estimates of lunar paleointensity based on whole rock samples, using both thermal and non-thermal paleointensity methods. That paper revealed evidence that SCP results on 5 samples ranging in age from 3.2- 3.9 Ga did not record a dynamo field despite apparent capacity to do so. A very young sample (~ 2 Ma) does retain a remanence but this is interpreted as a result of an impact not a native dynamo record. Supplementary material in the current manuscript indicates the existence of 3 samples (Apollo 70019, 15498, and 64455) with evidence for impact magnetization.

The new manuscript is based on the comparison of SCP and whole rock values from the 2 youngest of the 3.7-4.36 Ga samples, and these are the basalts, 70035 and 75035. The SCP produces a null magnetic field, and the high whole rock results are considered inconsistent and unrealistic. On this basis they accept all 4 of their null SCP results as affirmative evidence that any lunar dynamo was confined to the 140 My interval prior to 4.36 Ga. They speculate that absence of a lunar magnetosphere would allow transfer of Earth's Hadean magnetosphere to Moon and an associated record in the deeply buried lunar regolith. Geochemists might find this an interesting target for future study.

As I see it the current paper provides two stated advances over the 2021 work. Firstly, a direct comparison is made between SCP and whole rock paleointensity records on the same sample, reaffirming enthusiasm for the still challenging SCP experiments - the single crystals do have very weak magnetic signals. Secondly, the age of any putative dynamo is extended to that of the older samples, a 3.97 Ga breccia (61016) and 4.36 Ga anorthosite (60025). (No whole rock comparison is made on these samples.) The ages of these samples are cited as based on results from references 27 and 28. I have no expertise to judge the actual reliability of the $3.97 \pm .25$ GA and $4.360 \pm .003$ Ga age assignments, but the assertion of a dynamo lasting no longer than 140 My is clearly completely dependent on the second of these results - and perhaps on current views of age of Moon. Is 4.53 Ga still a good estimate (<https://eos.org/articles/the-moon-is-even-older-than-scientists-thought>)? I appreciate that the ages have been recalibrated with new decay constants, but could there be better determinations with more recent techniques? I found the manuscript lacking in detail about general reliability of these ages .

Overall this is an interesting study. It builds on SCP evidence supporting previous results that the lunar dynamo history is less complex than initially thought, and could be made significant with a more nuanced discussion of the reliability of the oldest age constraints.

Minor points:

Should Moon be capitalized as a proper noun when it is preceded by “the”?

Define REM’.